# Antibody–Drug Conjugates: The Dynamic Evolution from Conventional to Next-Generation Constructs

**DOI:** 10.3390/cancers16020447

**Published:** 2024-01-20

**Authors:** Virginia Metrangolo, Lars H. Engelholm

**Affiliations:** 1The Finsen Laboratory, Rigshospitalet, DK-2200 Copenhagen, Denmark; virginia.metrangolo@finsenlab.dk; 2Biotech Research & Innovation Centre (BRIC), Department of Health and Medical Sciences, University of Copenhagen, DK-2200 Copenhagen, Denmark

**Keywords:** antibody–drug conjugates, antibodies, targeted therapy, cancer immunotherapy, internalization, engineering

## Abstract

**Simple Summary:**

Since the first antibody–drug conjugate (ADC) was approved in 2000, the landscape of these targeted drugs has evolved over the years, profoundly changing the treatment scenario of various advanced-stage cancers, including solid tumors. With a better understanding of the construct’s mechanism of action and tumor biology, continuous efforts are made to fine-tune the ADC design and enhance the therapeutic index and tolerability of next-generation conjugates, promising to become the future of personalized cancer medicines. In this review, we outline the evolution of ADC development and discuss the existing challenges and future trends in the field.

**Abstract:**

Introduced almost two decades ago, ADCs have marked a breakthrough in the targeted therapy era, providing clinical benefits to many cancer patients. While the inherent complexity of this class of drugs has challenged their development and broad application, the experience gained from years of trials and errors and recent advances in construct design and delivery have led to an increased number of ADCs approved or in late clinical development in only five years. Target and payload diversification, along with novel conjugation and linker technologies, are at the forefront of next-generation ADC development, renewing hopes to broaden the scope of these targeted drugs to difficult-to-treat cancers and beyond. This review highlights recent trends in the ADC field, focusing on construct design and mechanism of action and their implications on ADCs’ therapeutic profile. The evolution from conventional to innovative ADC formats will be illustrated, along with some of the current hurdles, including toxicity and drug resistance. Future directions to improve the design of next-generation ADCs will also be presented.

## 1. Introduction

Antibody–drug conjugates are an actively growing class of precision immunotherapy that has reached clinical and regulatory milestones. Currently, there are fourteen FDA-approved ADCs available on the global market (as of December 2023), with hundreds more being evaluated in advanced clinical and preclinical trials [1,2,3,4]. ADCs are designed as pro-drugs consisting of a tumor-selective monoclonal antibody (mAb) covalently linked to a highly potent cytotoxic payload (also referred to as a warhead). As such, they act as “magic bullets” that specifically target cancer (or stromal) cells overexpressing the desired surface antigen (Ag)/receptor, enabling the selective delivery of the cytotoxic cargo and leading to targeted tumor eradication. This “Trojan horse” approach theoretically widens the therapeutic index of the cytotoxic agent by minimizing systemic drug exposure and dose-limiting toxicities to off-target healthy tissues, compared to standard chemo- and radiotherapies [1,2,3,4]. Furthermore, this approach may provide a superior antitumor effect than classical functional antagonism by unconjugated mAbs or small molecules, which have not proven to have compelling efficacy in some tumor types, as they would only slow or inhibit, but not eradicate, cancer cells. As such, the specific target-related function or signaling network is not the primary concern when designing an ADC, as it is its tumor-selective overexpression and endocytic function, as discussed later.

Table 1 provides a list of the current FDA-approved ADCs along with their main features. It is evident that most initial approvals were granted for hematological malignancies, and HER2 remains the main established ADC target for solid tumors, particularly breast cancer (BC). However, significant advances in target diversification and ADC design in the past decade have led to a rapid increase in the oncological indications being addressed. Seven ADCs are available for various solid cancers other than BC, four of which have received accelerated approvals in the last five years. These drugs target Nectin-4, Tumor-associated calcium signal transducer 2 (Trop-2), Tissue Factor (TF), and folate receptor alpha (FRα) [5]. Typically, ADCs have been authorized as monotherapy in advanced/metastatic refractory or relapsed settings. Following their exploration in combinatorial regimens with various agents, their application has also been extended to earlier settings, with some of these combinations now approved as first-line therapy for some cancer types [1,2,3,4]. As precision cancer medicine gains traction, ADCs and combination regimens tailored to specific oncological indications are expected to replace conventional treatment in the near future.

This review provides a concise overview of key ADC molecular components and mechanism of action and their implications on ADC efficacy and safety. We briefly illustrate the evolution in construct design, emphasizing the recent advances and considerations for developing the next-generation conjugates. Finally, we discuss current hurdles and open questions, highlighting future directions in the field. For an exhaustive outline of approved and clinically investigated ADCs and their mechanism of action, the readers are referred to [2,4,6].

## 2. Key Components of ADCs and Mechanism of Action

As shown in Figure 1, an ADC consists of a tumor-homing mAb and a cytotoxic payload chemically joined through a third component, the linker. This is designed to ensure controlled drug release at the tumor site by preventing premature payload detachment in the bloodstream and subsequent off-target toxicity.

This design dictates the ADC mechanism of action (MOA) (Figure 2). After Ag binding on the cell surface, the Ag-ADC complex is internalized via receptor-mediated endocytosis and traffics into the lysosome. Here, the ADC is processed according to the physicochemical properties of the linker (as detailed below) and releases the cytotoxic warhead that, once into the cytoplasm, ultimately triggers cell death or apoptosis, generally via DNA or tubulin targeting [1,2,3,4]. Lipophilic payloads can also diffuse from the ADC-targeted cells into neighboring cells and kill them regardless of their target expression. This mechanism, known as bystander killing, not only enhances the ADCs cytotoxicity but also enables targeting tumors with heterogeneous target expression, thereby extending the therapeutic benefit to a broader patient population. Notably, ADCs bearing cleavable linkers can generate such bystander effect independently of direct antigen binding and internalization but following extracellular payload release via alternative mechanisms. These include linker reduction, pH-induced, or proteolytic cleavage [7,8,9,10]. This mechanism has contributed to expanding the ADC target landscape by relieving the constraints of high expression and efficient endocytosis required by conventionally internalizing ADCs. As such, non-internalizing ADCs directed toward tumor microenvironment (TME) components, namely, extracellular-matrix proteins, stroma, or neovasculature, are gaining attention as an opportunity to broaden the oncological indications that may benefit from ADC therapy, particularly stroma-dense tumors, as detailed in Section 3 [7,8].

In addition to the payload, the mAb component may contribute to the ADC’s antitumor activity through its natural effector functions [2] (Figure 2). These include inhibition of target receptor downstream signaling pathways (e.g., regulating cell proliferation, metastasis, and survival) via the Fab (fragment antigen binding) region or Fc (fragment crystallizable)-mediated killing mechanisms like antibody- or complement-dependent cellular cytotoxicity (ADCC and CDD, respectively) and antibody-dependent phagocytosis (ADP), which directly engage innate immune or complement effectors. From this perspective, novel trends in ADC optimization intentionally exploit or modulate these mAb natural functions via Fc engineering to fine-tune the therapeutic index [5,11,12]. For example, Fc enhancement has demonstrated its benefit with the approval of an Fc-engineered afucosylated mAb-based ADC, namely, brentuximab vedotin (Blenrep^®^) [5,13].

However, while these Fc-mediated processes potentially boost the ADCs’ antitumor effect, they may also adversely affect their safety profile by increasing healthy tissue exposure through nonspecific drug diffusion, Fc-mediated uptake by immune cells, or recycling via neonatal Fc (FcRn) receptors [1,2,4,5,14]. Various Fc-silencing strategies are being investigated to overcome this issue in next-generation ADCs, and future trials will reveal the actual therapeutic benefit of this approach [5,11].

With the efficacy and safety of ADCs being strictly dependent on their mechanism of action, each of the three structural components, along with the target surface antigen and conjugation method, must be carefully selected and harmonically balanced according to the biology of the specific tumor type.

## 3. Target Antigen Selection: A Balance between Expression and Internalization

According to the currently accepted dogma, the optimal ADC target should be a surface-exposed (or extracellular) Ag serving as the delivery address and binding anchor for the mAb-targeting vehicle [2,4,5,6]. The Ag expression should be predominant or, preferably, exclusive to the tumor tissue compared to the healthy counterpart to ensure a wide and specific therapeutic window. High and homogenous Ag expression are generally required for effective antitumor activity. However, the bystander effect or potent payloads with unconventional mechanisms of action can be leveraged in the case of heterogeneously expressed targets [2,4,5,6].

The minimum Ag expression threshold required for ADC efficacy varies based on different Ag-related factors, including the target epitope and uptake dynamics, as well as the specific ADC construct and therapeutic indication. While a target Ag density of approximately 10 000 receptor copies/cell or higher has been proposed as a minimum threshold for efficient ADC activity, based on quantitative studies performed on well-established targets, like HER2 [15], the exact cut-off values need to be empirically determined case-by-case [5,16].

Recent evidence challenges these long-held beliefs regarding the Ag expression profile and threshold. Enhertu^®^, for instance, has proven effective in patients with low HER2 expression [6,17]. Furthermore, the clinical efficacy of other approved ADCs, such as polatuzumab vedotin (Polivy™) targeting CD79b, does not seem to correlate with the Ag density [18]. These findings emphasize the importance of considering other Ag- or ADC-related factors, like internalization or the payload mechanism of action, in patient selection [5,16].

To ensure successful payload delivery to the desired cells, the target Ag should possess endocytic- and lysosomal-trafficking properties [18]. From this perspective, receptor recycling has been shown to impair lysosomal routing and thus ADC activity by preventing the lysosomal-mediated release of free payload into the cytoplasm [5,18].

While information on the exact endocytic pathways of current FDA-licensed ADCs is fragmentary, preclinical reports have shown that a cell-specific balance between Ag expression and uptake drives ADC efficiency, with each of these parameters mutually compensating for each other, like in the case of CD22 and HER2 [18]. Nonetheless, a defined Ag density threshold seems to be required for some receptors, which efficient endocytosis may not offset, as seen for CD79b [18].

Of note, another factor influencing Ag intracellular routing and endocytic efficiency is the mAb-binding epitope, with some epitopes being more favorable for endocytosis than others. Also, the antibody itself can promote uptake via receptor cross-linking. This approach has been successfully applied to improve uptake by poorly internalizing receptors, like HER2, both in preclinical and clinical settings [18,19,20]. Bispecific and biparatopic mAbs/ADCs, targeting two distinct antigens and epitopes on the same Ag, respectively, are the main strategies currently explored [6,21,22,23,24].

Although all approved ADCs target tumor-specific antigens, the increasing understanding of tumor biology determinants, particularly the multifaceted importance of the TME in driving tumor progression, has raised interest in non-cellular, stromal, and vascular targets in the tumor milieu. The rationale is to target cancer cells indirectly by depriving them of their stromal support and nutrient supply [8]. The development of non-internalizing ADCs with mechanisms of extracellular payload release, previously mentioned, is accelerating this target diversification. Additionally, as stromal cells exhibit higher genome stability than cancer cells, this approach may reduce acquired resistance mechanisms caused by mutations typically observed for approved agents, like changes in Ag expression or endocytic pathways, as detailed later [8].

## 4. Antibody: The Precision Guide in ADC Therapy

The antibody moiety of the ADC serves as the tumor-targeting vehicle. As such, it should display high Ag specificity, strong binding affinity, and efficient cellular uptake [2,5,25].

Most of the early developed ADCs were based on mAbs selected based on their receptor antagonistic properties or for radionuclide therapy, delivered from the cell surface, rather than on their internalization performance [18]. This explains the unsatisfactory or often controversial effects observed with the first approved conjugates, for example, targeting CD33 or HER2 [18,26]. With the lessons learned from the approved ADCs, it is now apparent that the internalization efficiency and subsequent lysosomal trafficking (in the case of internalizing ADCs) is as fundamental as the target tumor-selective expression since it equally influences the ADC dose-dependent PK profile [6,18,27]. Therefore, studying and measuring the mAb-induced internalization is a vital aspect that should run in parallel with traditional ADC design areas to optimize the dosing regimen and enhance the therapeutic effect.

In this perspective, while it was believed that high binding affinity translates to more rapid and efficient endocytosis, it is increasingly clear that it can instead hinder penetration into solid tumors [2,25]. Indeed, the so-called “binding site barrier” traps high-target-affinity ADCs in the surrounding tumor vasculature, preventing their distribution to the central tumoral areas. Moreover, despite the potential of mAbs to better permeate cancer than normal tissues due to the typical leaky tumor vasculature, their large size poses a challenge to efficient solid tumor treatment, which explains the prevalent early application and clinical success of ADCs in hematological malignancies [2,4,25]. To address this challenge, various smaller mAb formats such as Fabs, scFvs (single-chain variable fragments), or nanobodies are now possible thanks to the rapid progress of recombinant and engineering strategies, as extensively reviewed in [28]. Although no such small-format conjugates have reached market approval yet, the encouraging preclinical and preliminary clinical results hold promise for future implementation of these smaller formats in the clinical arsenal. In addition, several alternative non-IgG scaffolds are being developed and explored, including protein and peptide formats, small molecules, aptamers, and ultrasmall C’Dot–drug conjugates [3]. Due to their lower molecular weight and versatile chemistry, these constructs have the potential of reduced immunogenicity and off-target interactions and enhanced tumor penetration than conventional ADCs, albeit with a lower serum half-life. Among the peptides, two have received FDA approval, respectively, the radiolabeled somatostatin analog ^177^Lu-Dotatate (Lutathera) for the treatment of metastatic neuroendocrine tumors [29] and melphalan flufenamide (melflufen) for treating patients with relapsed or refractory multiple myeloma, which was, however, later withdrawn from the market [30].

An ideal mAb candidate should also preserve its circulating half-life and biological features after conjugation and display minimal immunogenicity. From this perspective, the advent of recombinant and phage display technologies has enabled the transition from mouse-derived mAbs, which had high failure rates in the early stages of ADC development, to less immunogenic fully humanized antibodies, which constitute the backbone of all approved ADCs, except for brentuximab vedotin (Blenrep^®^) based on a chimeric mAb [2,25].

As the principal component of plasma immunoglobulins, the IgG isotype, particularly the IgG1 subclass, is the most common in clinically approved and investigational ADC drugs. Indeed, while having a similar serum half-life (approx. 21 days) as IgG2 and IgG4 subclasses, IgG1 are more potent inducers of immune effectors functions such as ADCC, ADCP, and CDC. On the other hand, IgG2, IgG3, and IgG4 subtypes might be less optimal due to enhanced risk of aggregation, rapid serum clearance (half-life ~7 days), and unusual Fab-arm dynamic exchange, respectively [25].

Different factors should, therefore, be considered and balanced when designing the optimal mAb for use in ADCs, rendering it a challenging case-by-case task.

## 5. From Traditional to Novel: Diversifying the ADC Payload Landscape

The ADC warhead is the active cytotoxic player of the construct that, depending on the MOA, defines the potency of the ADC molecule and potential target indications. As only about 2% of an intravenously injected dose of an ADC reaches the tumor site and due to the relatively low accommodation capacity of mAbs, the cytotoxin should be highly potent at low concentrations. Hence, payloads with IC_50_ values in the low sub-nanomolar and picomolar ranges are typically employed, which, as such, would consequently be too toxic to be used on their own [2,4,6,13,25,31].

Two principal payload classes dominate marketed ADCs: tubulin inhibitors and DNA-damaging agents (Figure 3). The first ones, with potency in the nanomolar range, include auristatins (e.g., MMAE and MMAF) and maytansines (e.g., DM1 and DM4) that, by interfering with the tubulin-polymerization dynamics, induce mitotic arrest thus blocking rapid tumor proliferation. Auristatin-based agents dominate the ADC landscape, accounting for eight of the approved conjugates [2,4,6,13,25,31].

Far superior cytotoxicity (~picomolar IC_50_ values) characterizes the DNA-targeting payloads, which inhibit cell proliferation via irreversible DNA damage through various mechanisms [2,4,13,25,31]. These include DNA double-strand breakage in the case of calicheamicins; DNA alkylation for duocarmycins; DNA intercalating agents, such as topoisomerase I inhibitors (SN-38 (7-ethyl-10-hydroxycamptothecin) and DXd (exatecan derivates)); and DNA cross-linkers like pyrrolobenzodiazepines (PBD). Due to their greater potency and independence from a specific cell division cycle, these payloads are more potent than the antimitotic agents and effective against low-Ag expressing or slowly dividing cells, like stromal cells [2,4,6,13,25,31]. Accordingly, such potent payloads appear better suited for targeting solid tumors, which often present with heterogenous Ag expression. However, the harmful side effects resulting from irreparable DNA damage raise safety concerns and limit the widespread use of these agents. Currently, loncastuximab tesirine (Zynlonta^®^) is the only approved ADC drug bearing a PBD dimer. Further research and clinical trial investigation on optimal drug testing and dosing regimens, together with linker and payload optimization, will be required to safely incorporate these classes in novel ADCs [31].

Besides potency, other payload-related features affecting ADCs efficacy are (1) the number of payload molecules per ADC or drug–antibody ratio (DAR); (2) cell-specific sensitivity and resistance profiles to the payload, e.g., the presence of multi-drug resistance (MDR) efflux pumps that can expel the payloads from the targeted cells; and (3) payload physicochemical properties influencing ADCs PK, in vivo metabolism, and safety profiles, like hydrophobicity inducing faster clearance or net charge in the free form, facilitating passive diffusion into surrounding cells and bystander killing [6,31].

While conventional cytotoxins have been at the forefront of ADC development, increasing efforts are being made to diversify the payload arsenal with molecules bearing original MOAs or less potent to address tumor indications that do not respond to ADCs or develop resistance mechanisms [2,6,13,31]. As for target diversification, this evolution in the ADC payload landscape is taking advantage of the breakthroughs in ADC design and cancer-targeting strategies. Innovative linker technologies now allow for improved construct stability, bystander-killing effects, and high DARs [6,13,25,31,32]. Indeed, while DAR values have been usually maintained under four to ensure optimal therapeutic index and limit the negative impact of hydrophobicity-induced ADC aggregation (faster clearance and systemic toxicity), the combination of novel stable linkers and less-potent payloads permits higher DARs, thus augmenting the overall drug tumor exposure [6,13,31]. Prompted by these advances in linker technologies, the recent development of a moderately potent ADC payload family, the topoisomerase 1 (topo-1) inhibitors, has marked a turning point in payload selection with the approval of two highly loaded (DAR 8) topo-1-based ADCs since 2019, trastuzumab deruxtecan (Enhertu^®^) and sacituzumab govitecan (Trodelvy^®^) [13,31]. These successful examples demonstrate the potential of unconventional payloads to increase the target indications and patients benefiting from ADCs. By providing orthogonal means to target cancer cells, tumors with moderate/low-Ag expression or resistant phenotypes, hard to target with traditional warheads, can potentially be addressed. Moreover, the efficacy demonstrated by DXd as a payload has not only spurred further advancements in its development but also catalyzed the initiation of numerous clinical trials exploring its mode of action [33].

Among the recently emerged payload classes, besides topo-1 inhibitors, are various other inhibitors, for example, targeting topoisomerase 2, transcription or translation effectors, anti-apoptotic proteins, or PROTACs. Also novel are radioactive isotopes [29] and immune stimulants [1,2,6,13,31]. The latter aims to engage the immune system and potentiate the ADC antitumor effect, with stimulators of interferon genes (STING) or Toll-like receptor (TLR) agonists amid the main classes currently explored (Figure 3) [1,2,6,13,31]. Different immune-stimulating ADCs have been developed, which have shown promising and long-term immune-mediated antitumor efficacy in several preclinical solid tumor models, with some at a clinical stage [2,13,31]. Notably, immune-stimulating activities have also been documented for conventional payloads, like anthracyclines (Doxorubicin or the more potent derivatives like Nemorubicin or PNU-159682) or auristatins (MMAE), which have demonstrated preclinical and clinical synergistic antitumor activities when combined with immune-checkpoint inhibitors (ICIs) [34,35,36,37,38,39,40,41,42]. Multiple mechanisms appear to be involved, including, among others, the induction of immunogenic cell death (ICD), dendritic cell activation, and increase in T lymphocyte infiltration, along with enhancement of immunological memory and expression of immune-regulatory proteins like programmed death ligand (PD-L)1 and MHC, as recently reviewed in [39,41]. While the ongoing clinical investigation will shed light on the actual clinical feasibility, therapeutic index, and benefits of the various emerging ADCs/ITs combinations over the standard of care and guide future trial design, the early encouraging results support optimism that they may revolutionize the treatment of immune refractory solid tumors, like pancreatic cancer, where IT modalities have yet to demonstrate clinical impact [34,35,36,37,38,39,40,41,42].

## 6. The Linker—A Balancing Bridge

The linker, tethering the cytotoxic payload to the mAb, is another crucial factor influencing ADCs therapeutic index and PK, as it determines plasma stability and release profile of the payload [2,4,25,32]. Generally, most ADC drugs incorporate two types of covalent linkers, cleavable and non-cleavable, which differ in their intracellular processing and systemic stability. While both have been shown to be safe in preclinical and clinical settings, cleavable linkers currently dominate the ADC landscape. For a comprehensive overview of the topic, the readers are advised to refer to [2,25,32,43,44].

Cleavable linkers are designed to be processed at the tumor site, taking advantage of the unique properties of the cancer TME over healthy tissues or systemic circulation [2,25,32,43,44]. They can be either chemically or enzymatically labile. Chemical linkers include hydrazone- or disulfide-bond-based linkers. The first type, used in commercially available Mylotarg^®^ and Besponsa^®^, is sensitive to low pH and can be hydrolyzed within acidic early endosomes post-uptake. The second one can undergo reduction via intracellular thiols such as glutathione (GSH), whose levels are generally elevated in cancer cells (1–10 mmol/L) than in blood (5 µmol/L) [45]. These linker types generate a membrane-permeable neutral payload able to promote bystander killing [1,2,32]. Notably, acid-sensitive linkers are insufficiently stable and can occasionally be hydrolyzed in the plasma, leading to premature drug release and off-target toxicity. This liability is one of the causes leading to the voluntary Mylotarg^®^ withdrawal by Pfizer from the US market in 2000, following the severe liver toxicity seen in patients [46,47,48]. The subsequent linker redesign and dosing schedule optimization have led to ADC reapproval in 2017 [46,49,50].

Conceived as an alternative strategy to chemically labile linkers, enzyme-cleavable linkers have reached clinical success in precise drug release [2,25,32,43,44]. They usually consist of dipeptides cleavable by lysosomal proteases, such as cathepsins, typically abundant in cancer cells. Nine of the approved ADCs include enzyme-labile linkers, mostly based on the well-established cathepsin-sensitive valine-citrulline (Val-Cit) dipeptide, used to construct the chimeric anti-CD30 antibody-MMAE conjugate, or brentuximab vedotin (Adcetris^®^). Alternative cathepsin-responsive dipeptides usually employed in ADC design are alanine–alanine (Ala-Ala), phenylalanine–lysine (Phe–Lys), or valine–alanine (Val–Ala) sequences. Other common protease-activated linkers targeting lysosomal enzymes rather than cathepsins include β-Galactosidase- or β-glucuronidase cleavable linkers [2,25,32,43,44]. Usually, due to the bulky nature of the payload, a para-aminobenzyloxycarbonyl (PABC) self-immolating spacer is also included in the linker moiety to facilitate enzyme access to the cleavage site. This unit further increases linker stability and allows straightforward payload release thanks to its self-cleavage capacity. Accordingly, enzyme-sensitive linkers provide a more precise release of the drug in the tumor milieu while also offering enhanced systemic stability than the chemical labile type due to their inertia at physiological conditions and protection by serum protease inhibitors [2,25,32,43,44]. However, while the extracellular cleavage of protease-labile linkers, e.g., by TME enzymes, may contribute to amplifying the ADC-induced anticancer effect by facilitating the above-stated bystander effect, it can also increase dose-limiting adverse effects (e.g., myelosuppression) due to premature drug release by blood cell proteases, like neutrophil elastases, as seen for Val-Cit-PABC linkers [51]. Furthermore, while Val-containing linkers are generally reasonably stable in human plasma, they have been shown to be susceptible to carboxylesterase 1C (Ces1C) in mouse and rat plasma [43,52]. Therefore, to allow for preclinical ADC evaluation in rodent models, considerable research efforts have been directed toward designing linkers with superior mouse plasma stability, either by structural optimization or by developing novel cleavable linker structures, as detailed below and thoroughly reviewed in [53].

Compared to cleavable linkers, the non-cleavable types, including thioether or maleimidocaproyl groups, are devised to release the cytotoxic payload only after complete lysosomal degradation. Accordingly, instead of the neutral payload, a “complex” of the drug linked to an amino acid residue of an antibody degradation product is released upon processing [2,25,32,43,44]. Due to their resistance to chemical or enzymatic hydrolysis, these linkers enjoy the advantages of increased plasma stability, longer half-lives, and lower off-target toxicity than the cleavable counterpart, rendering them potentially superior for targeting homogenously expressed Ag [2,25,32,43,44]. Nonetheless, as most tumors display heterogenous Ag expression, constructs incorporating cleavable linkers with membrane-permeant payloads are generally preferred due to their ability to induce bystander effects, unlike the non-cleavable linker type [2,25,32,43,44].

As the ADC field evolves dynamically and the payload arsenal rapidly expands, linker optimization has become crucial to maximizing the therapeutic index while improving biodistribution and PK profiles [1,6,25,32,43]. The currently explored strategies include (1) increasing the linker hydrophilicity (e.g., by incorporating negatively charged groups, like sulfonate, polyethylene glycol (PEG), phosphate, or pyrophosphate groups) to enhance systemic stability by reducing payload hydrophobicity-driven ADC aggregation and clearance [43,44]; (2) polyvalent or branched hydrophilic linkers, such as Fleximer^TM^ linkers or PEG chain additions, that enable high DAR without compromising the ADC physicochemical properties and PK [6]; and (3) tandem or dual-cleavage linkers requiring successive cleavage by lysosomal enzymes, which ensure tumor specificity while increasing both stability and tolerability [6,25,32]. Other recently emerged cleavable linker classes with higher plasma stability include lysosomal protease-cleavable linkers such as sulfatase- and legumain-cleavable linkers [53,54], while more innovative, albeit still exploratory, linker technologies include photo-sensitive and biorthogonal cleavable linkers, which widen up the opportunity for nonendocytic ADCs [32,43]. Paralleling the efforts in improving ADCs pharmacology is the expanding development of site-specific linkers and conjugation technologies, which allow for homogeneous constructs with superior therapeutic indices [32,43], as described in the following section.

## 7. Conjugation Technology

The bioconjugation method, which allows the joining of the ADCs’ structural backbones, is the cornerstone of the ADC technology. Traditional stochastic conjugation on the antibody cysteine (thiol) or lysine (ε-amino) residue side chains (via maleimide or amide coupling) is the most applied method. Considering the number of available residues typically found on a mAb (80–90 for lysine, 40 of which are generally modifiable, and 8 total disulfide bonds), the random payload coupling typically generates variable DARs (0–8) [2,13] (Figure 4). While such chemistries enjoy the advantages of relatively easy and fast reaction kinetics, the inherent heterogeneity of the resulting ADC mixtures causes variable therapeutic indices, PK, and stability profiles. Furthermore, one of the critical drawbacks of maleimide-based conjugates is their susceptibility to retro-Michael deconjugation and premature payload release in the presence of blood thiols [1,32,43,44]. These shortcomings have been partly responsible for clinical failures and discontinuation of some agents [4]. Therefore, intense research to address these issues has propelled the development of site-specific conjugation strategies to obtain homogeneous ADCs with predefined and consistent DARs (Figure 4). An extensive overview of the existing approaches is provided in [2,6,44,55,56,57].

Essentially, these involve orthogonal-coupling methods enabling the incorporation of unique anchor points for conjugation. These can either be engineered natural (e.g., cysteine residues used in the ThioMab™ technology) or non-canonical amino acids (e.g., ketones, azides, cyclopropenes or dienes) for chemical attachment or specific consensus sequences for enzyme-assisted ligation—for example, using bacterial transglutaminase (TG), formyl glycine-generating enzyme (FGE), S. aureus sortase, tyrosinase, and, more recently, ADP-ribosyl cyclases [2,13,44,55,56,57,58,59,60]. Another emerging (chemo)enzymatic approach is glycoconjugation, which harnesses the conserved Asparagine (Asn)297-linked glycosylation sites in the mAb Fc region for payloads coupling [2,6,13,44,55,56,57,58,59,60,61,62]. The distant localization of the glycan moieties and their different chemical composition compared to the mAb peptide chains enables site-specific coupling without compromising Ag binding. Among the existing methods are glycan metabolic engineering, glycosyltransferase-mediated addition of terminal sialic acid followed by oxidation, incorporation of azido- or keto-functionalized galactose, endoglycosidase-induced glycan remodeling, and incorporation of an azide anchor allowing copper-free click-coupling [44,57]. The latter approach developed by van Geel and co-workers (the GlycoConnect™ technology) was demonstrated to generate homogeneous, stable, and highly effective ADCs, outperforming the marketed Kadcyla^®^ in preclinical studies [63]. Another advantage of this method is the concomitant prevention or reduction in Fc-mediated off-target toxicity, which further improves ADC PK and tolerability [64], as previously mentioned.

Overall, site-specific ADCs and other bioconjugates have shown promising preclinical activities and higher therapeutic indices, namely, enhanced tumor uptake, improved safety and PK, and efficiency compared to conventionally coupled agents [2,6,44,55,56,57]. Moreover, the improved reproducibility of batch production renders both the manufacturing process and, to a certain extent, the therapeutic activity predictable, allowing for a tighter control of different variables. It is, therefore, not unlikely to envisage that such novel technologies, supported by the broad-spectrum advances in ADC development, will launch the next generation of potentially impactful ADCs in targeted anticancer therapy.

## 8. Existing Challenges and Opportunities of Next-Generation ADCs

Despite the upscaling clinical success of newly developed ADCs in the last decade, there remain different challenges in their widespread use as anticancer therapeutics and beyond. These include undesired AE, insufficient tumor penetration, complex PK, and drug resistance [2,5,16].

### Toxicity and Drug Resistance

Dose-limiting (DL) hematological toxicities, especially thrombocytopenia and neutropenia, are among the most severe AE commonly reported for the approved ADCs [5,16,65]. These are primarily attributed to Ag-independent off-tumor targeting, which can result from various mechanisms. These can be either non-specific uptake by Fc receptor-expressing cells (e.g., platelets) or via macro- or micro-pinocytosis processes, or eventually, premature payload release into the systemic circulation due to unstable linkers. As a result, highly perfused organs, like the bone marrow, alongside the liver, kidney, spleen, or gastrointestinal tract, with high cell renewal, can be exposed to ADC-induced toxicity, as for standard chemotherapy [2,5,6,16].

In some cases, the observed AE seems not correlated with the payload type or MOA, as within-class differences in toxicity profiles have also been described. For instance, auristatin MMAF causes ocular toxicities, while MMAE, which belongs to the same auristatin class, does not. Interestingly, this phenomenon has been reported for MMAF constructs bearing non-cleavable linkers. The charged payload complex that generates upon linker processing cannot diffuse out of the corneal epithelia, unlike the lipophilic MMAE, which is typically released from cleavable linkers. This evidence suggests that certain drug–linker combinations may contribute to specific off-target AE by altering the physicochemical properties of the construct [2,5,6,16].

Occasionally, regardless of the payload, on-target toxicities can also occur and relate to the healthy tissue expression of the target Ag. Indeed, most clinical and preclinical ADC targets are tumor-associated rather than tumor-selective [2,5,6,16]. However, this is not a “one-fits-all” situation, as some ADC targets with appreciable expression in non-malignant tissue, like Trop-2 or HER2, give no AE at those sites. A potential reason may be insufficient Ag expression or its spatial sequestration or distribution on the surface, limiting binding accessibility. Further complicating this scenario is the elusive occurrence of different toxicity patterns by the same ADCs based on the target tumor type [2,5,6,16].

Besides expression, other target-related attributes like altered endocytic and intracellular trafficking/recycling dynamics can induce toxicity, although information on these aspects remains vague for most investigated targets [2,5,6,16].

All these entangled factors render it difficult to foresee ADC’s safety profile and tolerability solely based on their structure and underline the relevance of careful dosing, AE monitoring and reporting, and appropriate intervention during clinical trials to shed light on potential toxicity mechanisms of ADCs [22]. The increasing application of ADCs and knowledge from clinical trials will guide researchers and clinicians in improving construct design and hopefully extend the population of cancer patients benefitting from next-generation ADCs.

As for any genre of therapy, another hindrance to ADC development is the development of drug resistance [1,2,16,50]. Based on the existing preclinical evidence, acquired resistance to ADCs appears far more convoluted and multifaceted than common drug-escape mutations (e.g., as for tyrosine-kinase inhibitors), probably mirroring the mechanistic complexities of this class of drugs [1,2,16,50]. Three main processes have been proposed: decrease in antigen expression following long-term exposure to ADCs, as seen for HER2-targeted T-DM1; alterations of intracellular routing pathways; and payload resistance. The latter generally involves the upregulation of efflux pumps, like ATP-binding cassette (ABC) transporters, typically observed for tubulin-inhibiting payloads (e.g., MMAE and DM1) and calicheamicin [1,2,16,50].

Further investigation and clinical evidence, currently limited, are essential to validate these assumptions in human patients. In addition to providing valuable insights for future drug optimization, this information will also facilitate the development of predictive biomarkers of therapeutic efficacy to direct patient stratification and drug regimens.

Current efforts to address the resistance phenomenon explore the use of dual-targeting modalities with bispecific antibodies, including cancer-stromal targeting agents or dual-payload constructs incorporating synergistic payloads with orthogonal MOAs [6,66]. By accurately proportioning the agent’s ratio and resulting ADC dosing, higher potency can be achieved while counteracting the incidence of resistant clones.

## 9. Conclusions

Since the first ADC approval in 2000, the ADC field has witnessed a dynamic evolution which, through trials and errors and continuous technology advances, aims to fine-tune this targeted drug modality, maximizing their pharmacological properties and therapeutic indices, and thus outcomes. The increased approvals in various solid tumors following target expansion and technology diversification highlight the importance of tailoring the drug construct according to tumor and target biology, offering a concrete opportunity for personalized cancer treatment. While PK, toxicity, and resistance mechanisms remain the main current challenges, the novel trends in payload expansion, linkers, and site-specific conjugation technologies embark on a journey of progress to develop more efficient and safer next-generation ADCs. The parallel development of immune-stimulating and stromal-targeting ADCs and combinations with immuno-oncology drugs promise to tackle difficult-to-treat cancers and provide a clinical benefit over unsatisfying standard-of-care approaches. Preclinical studies in relevant mouse models, accurate safety monitoring in clinical trials, and the development of predictive biomarkers will help refine drug construct development and select the most promising ADC combinations to advance into the clinics. The success of Enhertu^®^ in low-HER2-expressing cancer patients underscores the high potential and complexities of this evolving class of drugs. While there remains a need to develop strategic means for matching each ADC construct with the most appropriate cancer type and patient population, the potential of future ADCs to address multiple cancer indications renews hope in this research area.

## Figures and Tables

**Figure 1 cancers-16-00447-f001:**
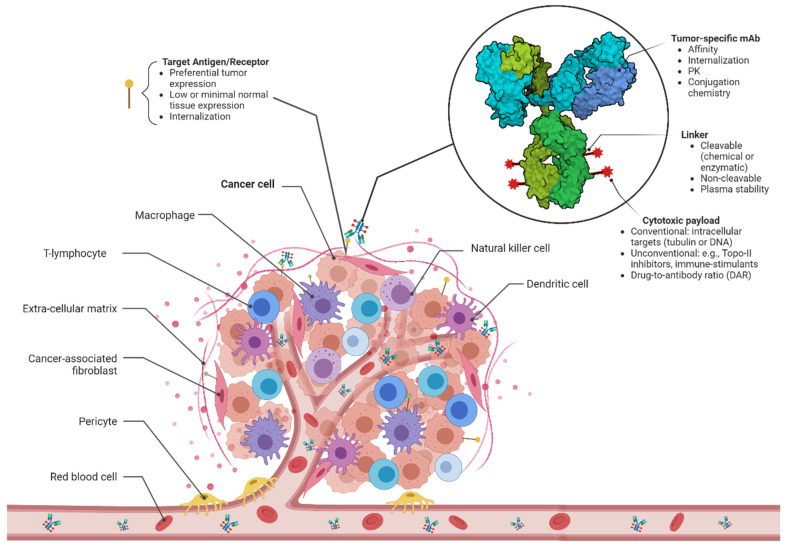
ADCs: the “magic bullets” for cancer therapy. An ADC is a bioconjugate of three structural backbones: a tumor-selective monoclonal antibody, a highly potent cytotoxic payload, and a stable cross-linker joining the two active components. The mAb acts as a targeted drug vehicle, enabling selective cytotoxin delivery into the target overexpressing cells at the tumor site, where, upon release, the toxin triggers tumor cell death through various mechanisms. This controlled drug release at the tumor site is governed by the linker, designed to be stable in the bloodstream and cleaved within the cancer cells or extracellularly in the tumor milieu. The properties of each ADC component and target antigen are highlighted. PK, pharmacokinetics. Created with BioRender.com (accessed on day 12 November 2023).

**Figure 2 cancers-16-00447-f002:**
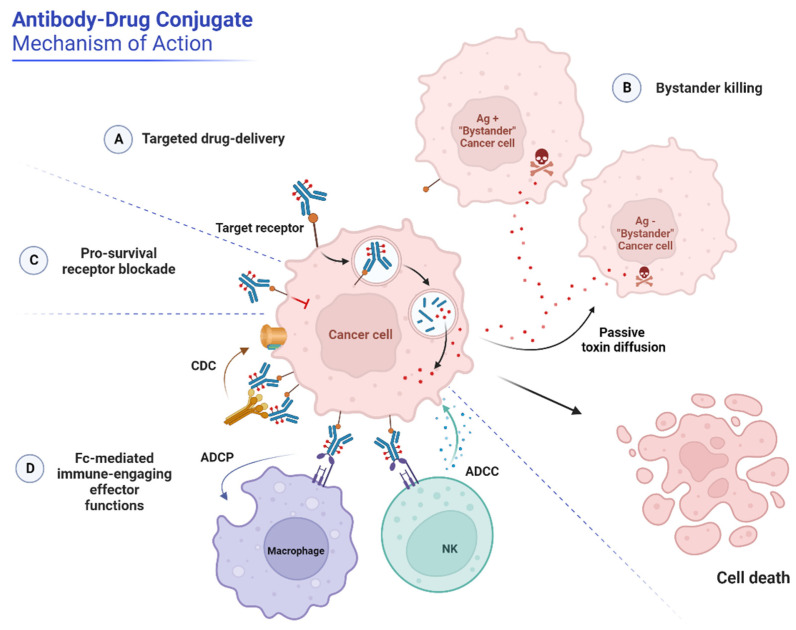
The mechanism of action of ADCs. (**A**) The primary MOA of ADCs involves targeted-drug delivery and tumor cell eradication following intracellular ADC processing and toxin release. (**B**) Lipophilic permeable payloads may passively diffuse out of the ADC-targeted cell into surrounding cells, leading to Ag-independent cell death or “bystander killing”. (**C**,**D**) The mAb component of ADCs may retain its activity profile, namely, antagonist functions, and interfere with the Ag downstream signaling to arrest cell growth or engage in Fc-mediated interactions with immune cell players {e.g., macrophages, natural killer (NK) cells, complement components), triggering antitumor immunity via ADCP, ADCC, or CDC effects. NK, natural killer cell. Created with BioRender.com (accessed on 6 November 2023).

**Figure 3 cancers-16-00447-f003:**
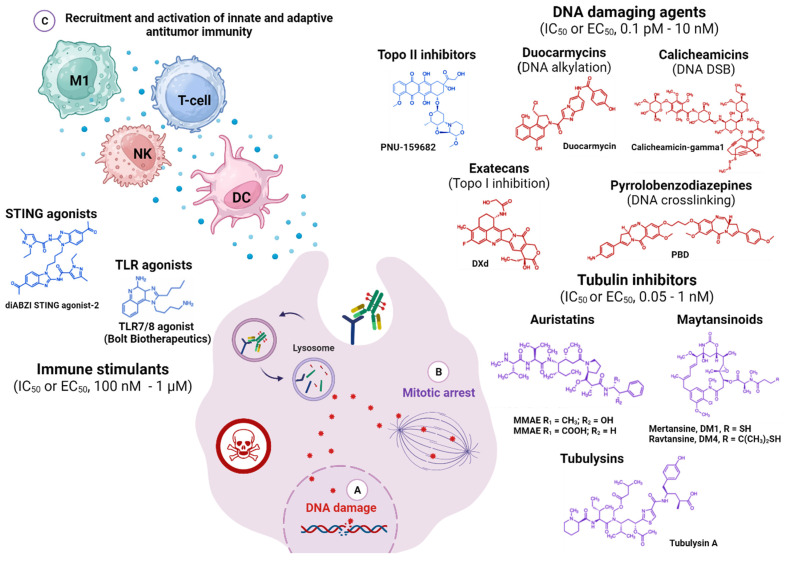
Representative classes of traditional and novel ADC payloads. The mechanism of action and therapeutic index of ADCs are largely dictated by the carried cytotoxic warhead. Based on their intracellular targets, payloads can work as tubulin inhibitors (shown in violet) or DNA-damaging agents (shown in red), which generally display higher potency. In blue are examples of unconventional agents recently explored in the ADC payload landscape: topoisomerase II (Topo-II) inhibitors and immune stimulants. The first work is on a complex and not fully unveiled mechanism involving not only Topo-II inhibition but also DNA intercalation, induction of ROS, and mitochondrial disruption. Immune stimulants function by inducing inflammatory cytokines that recruit and activate the host immunity, which finally mediates tumor cell killing. Immune-stimulating properties have also been described for the other two payload classes, including MMAE or anthracyclines, like PNU-159682. M1, type-1 macrophage; DC, dendritic cell. Created with BioRender.com (accessed on 12 November 2023).

**Figure 4 cancers-16-00447-f004:**
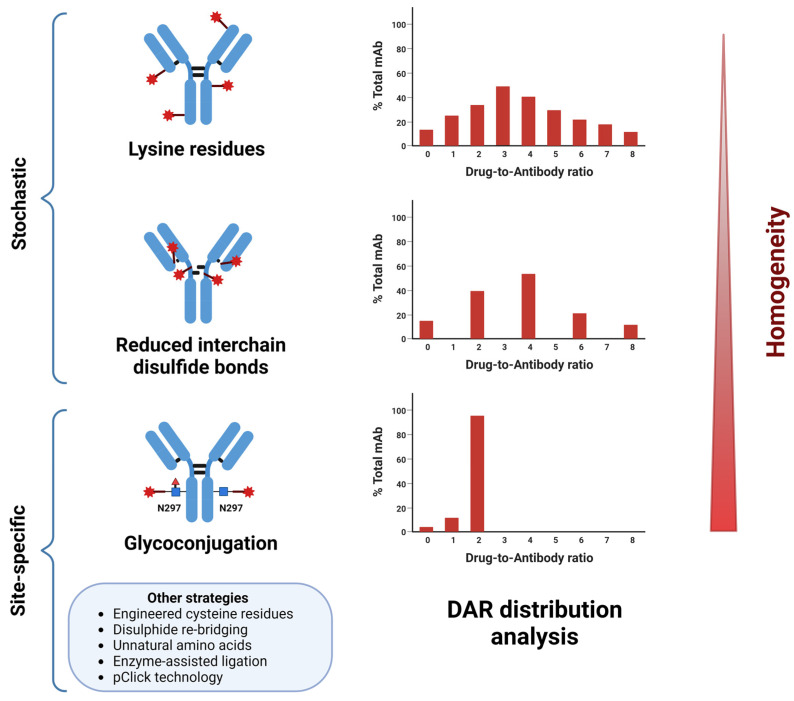
Conjugation strategies for ADC development. ADCs can be produced by traditional stochastic coupling on exposed lysine or reduced interchain cysteine residues or via site-specific conjugation approaches, like glycoconjugation. The latter yields more homogeneous ADCs in terms of DAR and conjugation site than randomly linked ADCs. Lysine and cysteine coupling results in a DAR of 0–8 and potential conjugation at 40 and 8 residues/mAb, respectively. Glycoconjugation at the two conserved N297-linked glycans produces DAR 2 ADC. Other site-specific conjugation strategies, such as cysteine engineering, enable us to double the DAR. Created with BioRender.com (accessed on 27 November 2023).

**Table 1 cancers-16-00447-t001:** Summary of clinically approved ADCs worldwide and their common adverse events.

ADC (Company)	Trade Name	Target Ag	Linker	Payload	Average DAR	Approved Date	Approved Indications/ Drug Regimen	Common AEs
**Hematological Malignancies**								
Gemtuzumab ozogamicin (Pfizer, New York, NY, USA)	Mylotarg^®^	CD33	Cleavable (hydrazone)	calicheamicin	2–3	2000/5, (USA; withdrawn from the market in 2010); 2017/9 (USA), 2018/4 (EU)	Newly diagnosed AML (including pediatric patients)	Normal tissue expression of the Ag: VOD, hemorrhage, hepatotoxicity Payload-related: hepatic dysfunction, myelosuppression Other AEs: fatigue, pyrexia, nausea, vomiting, headache, infection, stomatitis, diarrhea, abdominal pain
Brentuximab Vedotin (Seagen, Bothell, WA, USA)	Adcetris^®^	CD30	Cleavable (mc-VC-PABC)	MMAE	4	2011/8 (USA); 2012/10 (EU)	R/R CD30-positive HL and systemic ALCL, including some types of PTCL and previously untreated stage III or IV cHL; in combination with chemotherapy	Payload-related: peripheral sensory neuropathy, myelosuppression Other Aes: upper respiratory tract nausea, fatigue, diarrhea, pyrexia, vomiting, arthralgia, pruritus, myalgia, alopecia
Inotuzumab ozogamicin (Pfizer)	Besponsa^®^	CD22	Cleavable (hydrazone)	calicheamicin	5–7	2017/6 (EU); 2017/8 (USA)	R/R B-cell precursor ALL	Payload-related: hepatic dysfunction, myelosuppression Other Aes: hemorrhage, pyrexia, nausea, infection, headache
Moxetumomab Pasudotox (AstraZeneca, Cambridge, UK)	Lumoxiti^®^	CD22	Cleavable (mc-VC-PABC)	PE38	NA	2018/9 (USA; withdrawn in 2023/7); 2021/2 (EU, 2021/7)	R/R HCL who have failed to receive at least two systemic therapies	Body swelling, nausea, fatigue, headache, fever, constipation, anemia, diarrhea, capillary leak syndrome, and hemolytic uremic syndrome
Polatuzumab vedotin (Roche, Basilea, Switzerland)	Polivy^®^	CD79B	Cleavable (mc-VC-PABC)	MMAE	3.5	2019/6 (USA); 2020/1 (EU)	R/R DLBCL, after at least two prior therapies. In combination with bendamustine plus rituximab	Payload-related: peripheral sensory neuropathy, myelosuppression
Belantamab mafodotin (GSK, London, UK)	Blenrep^®^	BCMA	Non-cleavable (mc)	MMAF	4	2020/8 (USA; terminated in 2022/11); 2020/8 (EU; terminated in 2023/9)	R/R MM, after at least four treatments, including anti-CD38 mAbs, proteasome inhibitors, and immunomodulators	Payload-related: ocular toxicity Other AEs: myelosuppression, pyrexia, nausea, increased aspartate aminotransferase, keratopathy
Loncastuximab tesirine (ADC Therapeutics, Épalinges, Switzerland)	Zynlonta^®^	CD19	Cleavable (dipeptide)	PBD dimer (SG3199)	2.3	2021/4 (USA); 2022/12 (EU)	R/R large B-cell lymphoma after two or more lines of systemic therapy (adults)	Payload-related: increased gamma-glutamyl transferase, fluid retention, myelosuppression Other AEs: hyperglycemia, transaminase increase, hypoalbuminemia, musculoskeletal pain, fatigue
**Solid cancers**								
Ado-trastuzumab emtansine (Roche)	Kadcyla^®^	HER2	Non-cleavable (SMCC)	DM1	3.5	2013/2 (USA); 2013/11 (EU)	Adjuvant treatment of patients with HER2-positive early breast cancer presenting residual invasive disease after neoadjuvant therapy	Normal tissue expression of antigen: cardiac toxicity Payload-related: myelosuppression, increased transaminases, peripheral sensory neuropathy Off-target toxicity: interstitial pneumonitis Other AEs: ocular toxicity, fatigue, nausea
Enfortumab vedotin (Seagen)	Padcev^®^	Nectin-4	Cleavable (mc-VC-PABC)	MMAE	3.8	2019/12 (USA); 2022/4 (EU)	Advanced or metastatic urothelial cancer patients previously treated with platinum chemotherapy and a PD-L1/PD-1 inhibitor	Normal tissue expression of antigen: dysgeusia Payload-related: peripheral sensory neuropathy Other AE: rash, alopecia, dry eyes and skin, pruritus, diarrhea, fatigue, alopecia, nausea, decreased appetite
Fam-trastuzumab deruxtecan (Daiichi Sankyo, Tokyo, Japan)	Enhertu^®^	HER2	Cleavable (tetrapept)	DXd	7–8	2019/12 (USA); 2021/1 (EU)	Unresectable or metastatic HER2-positive breast cancer patients after two or more prior HER2-targeting regiments; locally advanced or metastatic HER2-positive gastric or gastroesophageal junction adenocarcinoma patients after a prior trastuzumab-based regimen	Normal tissue expression of antigen: cardiac toxicity Payload-related: gastrointestinal toxicity, myelosuppression Off-target toxicity: interstitial pneumonitis, nausea, fatigue, alopecia, vomiting, decreased appetite, diarrhea, constipation
Sacituzumab govitecan (Immunomedics, Morris Plains, NJ, USA)	Trodelvy^®^	Trop-2	Cleavable (CL2A)	SN38	7.6	2020/4 (USA); 2021/11 (EU)	Unresectable locally advanced or metastatic TNBC patients who have received two or more systemic therapies (of which at least one is for metastatic disease)	Normal tissue expression of antigen: skin rash, hyperglycemia Payload-related: myelosuppression, diarrhea Other AEs: alopecia, vomiting, nausea, constipation
Cetuximab sarotalocan (Rakuten Medical, San Diego, CA, USA)	Akalux^®^	EGFR	NA	IRDye700DX	1.3–3.8	2019/9 (China)	Unresectable locally advanced or recurrent HNSCC	Application site-pain, local edema
Disitamab vedotin (RemeGen, Yantai, China)	Aidixi^®^	HER2	Cleavable (mc-VC-PABC)	MMAE	4	2021/6 (China)	Locally advanced or metastatic gastric cancer patients (including gastroesophageal junction adenocarcinoma) previously treated with at least 2 types of systemic chemotherapy	Myelosuppression, gastrointestinal diseases, fatigue, fever
Tisotumab vedotin (Genmab, Copenhagen, Denmark/Seagen)	Tivdak^®^	TF	Cleavable (mc-VC-PABC)	MMAE	4	2021/9 (USA)	Adult patients with metastatic or recurrent cervical cancer or after chemotherapy	Normal tissue expression of antigen: hemorrhagic complication and conjunctival reaction Payload-related: peripheral sensory neuropathy, myelosuppression
Mirvetuximab soravtansine (ImmunoGen, Waltham, MA, USA)	Elahere^®^	FR	Cleavable (Sulfo-SPDB)	DM4	3.5	2022/11 (USA)	Adult patients with folate receptor–alpha positive ovarian cancer, fallopian tube cancer, or primary peritoneal cancer refractory platinum-based chemotherapy or after 1 to 3 prior chemotherapies	Payload-related: peripheral neuropathy, myelosuppression Off-target toxicity: ocular toxicity Other AEs: reversible ocular toxicity (uveitis and keratopathy), pneumonitis

Abbreviations: AEs, adverse effects; ALCL, anaplastic large cell lymphoma; AML, acute myeloid leukemia; cHL, classical Hodgkin lymphoma; BCMA, B-cell maturation antigen; CL2A, a cleavable complicated PEG8- and triazole-containing; DAR, Drug-to-Antibody ratio; DLBCL, diffuse large B-cell lymphoma; DM, derivative of maytansine; DXd, Exatecan derivative for ADC; EGFR, epidermal growth factor receptor; GSK, GlaxoSmithKline Inc.; HCL; hairy cell leukemia; HL, Hodgkin lymphoma; HNSCC, head and neck squamous cell carcinoma; mc-VC-PABC, maleimidocaproyl-valine-citrulline-p-aminobenzoyloxycarbonyl; mc, maleimidocaproyl; MM, multiple myeloma; MMAE, monomethyl auristatin E; MMAF, monomethyl auristatin-F; NA, non-applicable; SMCC, succinimidyl-4-(N-maleimidomethyl)cyclohexane-1-carboxylate; PTCL, peripheral T-cell lymphomas; PABC-peptide-mc linker; PBD, pyrrolobenzodiazepine; PD-L1, programmed cell death-ligand 1; PD-1, programmed cell death protein-1; PE38, a 38kD fragment of Pseudomonas exotoxin A;R/R, relapsed or refractory; tetrapept, tetrapeptide; SN38, active metabolite of irinotecan; TF, tissue factor; TNBC, triple-negative breast cancer; VOD, veno-occlusive disease.

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
