# Peer review of "Antibody–Drug Conjugates: The Dynamic Evolution from Conventional to Next-Generation Constructs"

_cancers, 2024, doi:10.3390/cancers16020447_

Round 1

Reviewer 1 Report

Comments and Suggestions for Authors

Metrangolo et al. provide outline the ADC development and discuss the challenges and future trends. The manuscript has well been written.I did not consider there to be any major points to add to this paper.

The full spelling of “TME” does not seem to be available. Does it refer to the “tumor microenvironment”?

Figures 3 and 4 are not cited in the text. The authors should cite them in the text.

There was a small description of the concomitant use of ADC and immune checkpoint inhibitors (lines 299-305). I thought that the phenomenon of "immunogenic cell death (ref. 33, 34 and 39)" could have been mentioned.

Author Response

Thank you very much for the time you spent revising the manuscript and providing valuable feedback and comments.

Please, find the detailed responses below, while the corresponding revisions/corrections are highlighted in track changes in the re-submitted file.

Comment 1: the full spelling of “TME” does not seem to be available. Does it refer to the “tumor microenvironment”?

Response 1: Yes, the word was not explicated. It is now (line 117 in the text).

Comment 2: Figures 3 and 4 are not cited in the text. The authors should cite them in the text.

Response 2: Yes, thanks. We have now inserted them in the text on pages 11, 12, and 15. 

Comment 3: There was a small description of the concomitant use of ADC and immune checkpoint inhibitors (lines 299-305). I thought that the phenomenon of "immunogenic cell death (ref. 33, 34 and 39)" could have been mentioned.

Response 3: Thanks for the relevant input. We have now elaborated on the proposed mechanisms responsible for the synergism between ADCs and ITs, including immunogenic cell death (page 12).

Again thanks for your valuable comments.

Reviewer 2 Report

Comments and Suggestions for Authors

ADCs represent an innovative strategy that combines the precision of monoclonal antibodies with the potency of highly cytotoxic agents. By specifically delivering therapeutic payloads to tumor sites, ADCs have the potential to significantly mitigate side effects. As they increasingly play a pivotal role in first-line cancer treatments and secure regulatory approvals, advancements in production technology have resulted in a growing array of approved ADCs or those in advanced clinical trials. The expanding diversity of antigenic targets and bioactive payloads is swiftly broadening the spectrum of tumor indications for ADCs, making this review highly relevant to its readers. However, certain issues require attention before publication, as outlined below:

The widely utilized click reaction for synthesizing antibody-drug conjugates, along with the conjugation methods involving native functional groups such as SH, NH2, S-S etc, deserves a comprehensive discussion. This would enhance the understanding of the synthesis techniques employed in ADC development. The review should also emphasize more on the key factors influencing the efficacy of antibody-drug conjugates. A comprehensive examination of these factors will provide valuable insights into optimizing the performance and therapeutic potential of ADCs in cancer treatment. Addressing these aspects will contribute to the overall robustness and completeness of the review.

The incorporation of a discussion on the use of cleavable linkers is essential. Highlighting the role and significance of cleavable linkers would contribute to a more thorough exploration of the diverse strategies employed in ADC design.

Author Response

Thanks for taking the time to revise the manuscript and provide us with valuable feedback and inputs for improvement. 

Please, find the detailed responses below, while the corresponding revisions/corrections are highlighted in track changes in the re-submitted document.

Comment 1: The widely utilized click reaction for synthesizing antibody-drug conjugates, along with the conjugation methods involving native functional groups such as SH, NH2, S-S etc, deserves a comprehensive discussion. This would enhance the understanding of the synthesis techniques employed in ADC development.

Response 1: Thanks for the input. We have now expanded our discussion on site-specific conjugation technologies, ensuring to include essential and comprehensive references for a more in-depth understanding of the topic (pages 15-16)

Comment 2: The review should also emphasize more on the key factors influencing the efficacy of antibody-drug conjugates.

Response 2: Thanks for the comment. We have further elaborated our considerations on the crucial determinants of ADCs efficacy and safety. On page 9 we included considerations on the Antigen, on pages 11 and 12 we elaborated on the payloads, on pages 14-16 we expanded our discussion on linkers and conjugation methods.

Comment 3: The incorporation of a discussion on the use of cleavable linkers is essential. Highlighting the role and significance of cleavable linkers would contribute to a more thorough exploration of the diverse strategies employed in ADC design.

Response 3: Thanks for the valuable input. We have thoroughly revised the sections concerning cleavable linkers, describing their advantages and drawbacks and outlining recent research trends in their optimization.

Thanks again for your comments.